# Indications of alcohol or drug use disorders in five different national registers in Sweden: a cross-sectional population-based study

Andreas Lundin [ID],[1,2] Anna-Karin Danielsson [ID],[1] Christina Dalman,[1,2] Anna-Clara Hollander [ID] [1]

[1]Epidemiology of Psychiatric Conditions, Substance Use and Social Environment (EPiCSS), Department of Global Public Health, Karolinska Institutet, Stockholm, Sweden
[2]Centre for Epidemiology and Community Medicine, Region Stockholm, Stockholm, Sweden

**Correspondence to**
Dr Anna-Clara Hollander;
anna-clara.hollander@ki.se

## ABSTRACT

**Objective** The purpose of this study is to examine the prevalence of indications of alcohol or drug use disorders in five different national Swedish registers and to investigate the correlation between these registers. Furthermore, the intent is to investigate whether combining data from different registers increases the prevalence of these indications in the population due to the identification of different demographic groups in different registers.

**Design** Cross-sectional study.

**Setting and participants** Individuals living in Sweden aged 20–64 years in 2006, n=5 453 616.

**Primary outcome** National registers included the Registers of Inpatient Care, Outpatient Care, Medications, Social Insurance and Convictions. Demographic variables were sex, age, migrant status, education and civil status. Indications of alcohol or drug use disorders were presented as prevalence in percentage (%), correlation was examined using phi correlation coefficients and differences across demographic factors were studied using logistic regression.

**Results** The prevalence of an indication of alcohol or drug use disorder varied between registers, meaning that prevalence increased when all registers were considered together. The prevalence of alcohol use disorder increased by 60% and 66% among men and women, respectively, while the prevalence of drug use disorder increased by 45% and 80% among men and women, respectively, when all registers were combined, compared with only using the register with the highest prevalence. Registers contributed different indications of drug and alcohol use disorders.

**Conclusions** Accurate estimates of alcohol or drug use disorders are critical for healthcare and rehabilitation. This study shows that using a single register alone underestimates the burden of disease unevenly, while combining a range of registers can provide a more accurate picture.

## BACKGROUND

Alcohol and drug use cause a substantial disease burden globally, the composition and extent of which varies between countries.[1] Globally, in 2016, 4.2% of all disability-adjusted life years (DALYs) were attributable

---

## STRENGTHS AND LIMITATIONS OF THIS STUDY

⇒ The study has a large sample size as it included all individuals living in Sweden aged 20–64 years in 2006 (n=5 453 616).

⇒ The five national registers included in the study are of high quality.

⇒ A limitation of the study is that it examines the period prevalence of substance use disorder and does not take previous use into account.

⇒ The validity of indications of alcohol and drug use disorder depends on the validity of the different registers.

---

to alcohol use, 1.3% were attributed to drug use[1] and lost DALYs were due to risk of injury, premature death and other negative consequences.[2] However, the risk posed by using alcohol and drugs differs between men and women, by socioeconomic circumstance, and for migrants, by country of birth.[3–5] The term 'substance use disorder' refers to both substance abuse and substance dependence, and although studies of substance use disorder are common in population surveys,[3 6] registry-based studies provide an additional way to study substance use disorders.[7–9] There are many advantages to using survey data in studies of alcohol and drug use disorders in a population, such as the ability to study different degrees of the disorder, as the participants describe their consumption and their substance use related problems themselves. However, disadvantages of surveys include: decreasing response rate in population surveys in general[10]; selection bias, as some socioeconomic groups are less likely to respond to surveys than others; and the fact that heavy substance use can affect memory,[11] increasing the risk of recall bias.

In Scandinavia and Finland, healthcare visits are recorded in local and national administrative registers covering the entire

population. Such population-based registers have benefited psychiatric research immensely.[12] These health registers are generally considered valid,[13] but using registers collected for administrative purposes in research can give misleading results if there is a systematic underuse of care in specific subgroups. In Sweden, to date, studies of substance use disorder have used the National Patient Register,[4 7–9] assuming the perspective that the most severe cases of substance use disorder will require care. This register has been an important source of knowledge about the social determinants and consequences of severe alcohol and drug use disorders. However, it has long been recognised that—in addition to only detecting the most severe cases—these registers possibly underestimate severe cases in specific subgroups where underutilisation is pronounced.[14] As register-based research could be a powerful tool for population-based studies of substance use disorders, there have been attempts to expand the use of population registers by including additional registers with possibly less severe cases, such as registers with diagnoses from general practitioners' (GP) surgeries,[15] the Crime Prevention Council's register[16] and the Register of Prescribed and Purchased Medicine.[17] Studies using the additional registers recognise that appearing in the additional registers is not the same as being diagnosed with a substance use disorder, but acknowledge that it is a possible indication of a substance use disorder. However, despite being used in studies of the determinants and consequences of a substance use disorder, this practice has not yet been tested empirically.

The purpose of this study is to examine the prevalence of indications of alcohol or drug use disorders in five different national Swedish registers. This was done to investigate the extent of correlation between the registers to find out whether combining registers increases the prevalence of such indications in the population due to registers identifying different demographic groups such as different age groups, sexes, educational levels and migrant or marital status.

## METHODS

### Public and patient involvement
No patients were involved in setting the research question or the outcome measures, nor were they involved in developing plans for design or implementation of the study. No patients were asked to advise on interpretation or writing up of results. However, we will disseminate the results of our research to agencies responsible for the healthcare of refugee and migrant groups in Sweden.

### Design
We conducted a historical cross-sectional register-based study in several Swedish national registries. To link registers, we used the personal identity numbers that are uniquely assigned to every resident at birth or on immigration.

### Data sources
Data were extracted from a linked register database called Psychiatry Sweden (PS) held by the research group Epidemiology of Psychiatric Conditions, Substance Use and Social Environment. PS consists of several national and regional registers.

For the registers of alcohol and drug use disorders, we used three registers from the National Board of Health and Welfare: the Patient Register (which actually contains two registers: one for inpatient care (henceforth called the 'Inpatient Register') and one for specialised outpatient care (henceforth called the 'Outpatient Register')). Data on psychiatric inpatient care have been accessible in these registers since 1973, and there has been good coverage of psychiatric outpatient data since 2006.[18] The Register of Prescribed and Purchased Pharmaceuticals (*Läkemedelsregistret*) (accessible since 2006)[19] is a register of all medicines prescribed in healthcare and dispensed at pharmacies (henceforth called the 'Register of Medications'). We also included the Swedish Social Insurance Agency's (*Försäkringskassan*) *Mikrodata* for analysis of social insurance (henceforth called the 'Social Insurance' register) (accessible since 1992),[20] from which we used the variables 'sickness absence' and 'disability pension'. 'Sickness absence' refers to a temporarily reduced work capacity due to alcohol and drug use disorders. A disability pension is a public financial support provided to the individuals who permanently leave the labour market before the age of statutory retirement, usually preceded by sick leave. From the Crime Prevention Council's register of convicted criminals (*Brottsförebyggande Rådets* (BRÅs)) Register över lagförda personer[21] (since 2000), we used the register of prosecuted convictions based on the information registered in the Swedish Public Prosecutor's Office and courts (henceforth called the 'Register of Convictions').

In addition, we used registers from Statistics Sweden for demographic factors, including total population, the Longitudinal Integration Database for Health Insurance and Labour Market Studies (LISA) (since 1990)[20] and the Longitudinal Database for Integration Studies (STATIV) (since 1997).[22]

### Population
The population consisted of all persons in Sweden aged 20–64 years in 2006 and was identified in LISA. The lower age limit was determined by not having access to data from child and adolescent psychiatry from all over Sweden, and the higher age limit was determined as persons over 64 years cannot be included in the register of Social Insurance due to the Swedish age of retirement in 2006. The population were then followed up in various registers regarding indications of alcohol and drug use disorder between 2006 and 2016.

### Definitions of indication of alcohol or drug use disorder
The present study focused on use so extensive and problematic that it was captured in national registers and will

henceforth be called a 'register indication' of alcohol or drug use disorder. The registers primarily use the International Statistical Classification of Diseases and Related Health Problems, 10th Edition (ICD-10)[23] system of classification, where alcohol and drug use disorder has six possible criteria:

► Craving.
► Loss of control.
► Prioritisation of alcohol/drug consumption over other activities.
► Continued intake, despite adverse effects.
► Increased tolerance.
► Abstinence.

For a person to be diagnosed according to ICD-10, three of six criteria must have been met during the past year.[23]

### Alcohol use disorder

In both the patient registers and the social insurance register, an indication of alcohol use disorder was an ICD-10 diagnosis of 'mental disorders and behavioural disorders caused by alcohol (F10)'.

In the Register of Convictions, an indication of alcohol use disorder was defined as relevant alcohol-related crimes, such as drunk driving, aggravated drunk driving or care of intoxicated persons (Act (1951: 649) on penalties for certain traffic offences (§4 drunk driving and §4a aggravated drunk driving, respectively). Since one of the ICD-10 criteria for alcohol use disorders is still drinking despite negative effects, it was only counted as a register indication if an individual had more than two convictions in the same calendar year.

In the Register of Medications, an indication of alcohol use disorder was defined as having been prescribed medicine to treat alcohol use disorder, including disulfiram (ATC NO7BB01), acamprosate (ATC NO7BB03), naltrexone (ATC NO7BB04) or nalmefene (ACT NO7BB05).

### Drug use disorder

In the patient registers and the social insurance register, an indication of drug use disorder was an ICD-10 diagnosis of chapters F11–F14, F16 and F18–F19, which are mental disorders and behavioural disorders caused by:

► Opioids (F11).
► Cannabinoids (F12) under the heading "Drug use disorder".
► Sedatives or hypnotics (F13).
► Cocaine (F14).
► Hallucinogens (F16).
► Volatile solvents (F18).
► Psychoactive substances (F19).

In the Register of Convictions, an indication of drug use disorder was a relevant drug-related crime (Narcotics Penal Code (1968: 64) and the Act (1991: 1969)) on the prohibition of certain substances such as possession, use of or other engagement with drugs. As one of the ICD-10 criteria for drug use disorder is continued use despite negative effects, it only counted as a register indication when a person had more than two convictions during the same calendar year.

In the Register of Medications, an indication of drug use disorder was having been prescribed and dispensed buprenorphine N07BC01, methadone N07BC02, levomethadone N07BC05 or buprenorphine or combinations N07BC51 (only drugs for opiate dependence were available).

### Covariates

► Sex was defined as sex assigned at birth: male or female.
► Age was categorised into groups: 20–29, 30–39, 40–49, 50–59 and 60–64 years.
► Migration status was grouped into those born in Sweden, migrants and refugees; 'refugees' were defined as having received a residence permit for refugee reasons according to the Swedish Migration Agency.
► Education level was grouped according to Swedish education nomenclature as presecondary, upper secondary, postsecondary and unknown.
► Marital status was grouped as married (including registered partnership), divorced (including deregistered partnership), widow/widower and unmarried.

### Statistical analysis

Alcohol and drug use disorder in the registers were presented as cumulative period prevalence and reported in percentages. Correlation, that is, the extent to which those with an indication of substance use disorder in one register appear in other registers—was examined with phi correlation for dichotomous variables. Phi correlation is interpreted as a scale where the correlation coefficient is between 1 and –1, where 1 means maximum positive correlation, 0 is no correlation at all and –1 is maximum negative correlation. Relationships between the demographic variables (age, sex, migrant status, level of education and marital status) and indication of alcohol or drug use disorder in different registers were examined using logistic regression and presented as ORs and 95% CIs. All regression models were multivariable models including the variables age, sex, migrant status, education level and marital status.

### RESULTS

The population consisted of all individuals in Sweden aged 20–64 years in 2006, for a total of 5.45 million individuals, and is described in table 1.

Table 2 shows the prevalence of an indication of alcohol use disorder by sex and age in the population. The prevalence of individuals with a register indication for alcohol use disorder differed between registers. In total, 2.0% of the women and 4.2% of the men appeared in one of these registers. Due to the doubled prevalence of an indication among men, compared with women, table 2 is

**Table 1** Characterisation of the population in 2006 in number (n) and per cent (%)

| | | n (%) |
|---|---|---|
| Characteristics | **Total** | 5 453 616 (100) |
| Sex | Male | 2 781 344 (51) |
| | Female | 2 672 272 (49) |
| Migration status | Born in Sweden | 4 581 037 (84) |
| | Born abroad | 818 042 (15) |
| | Refugee | 109 072 (2) |
| Education | Preupper secondary school | 872 578 (16) |
| | Upper secondary | 2 617 736 (48) |
| | Postsecondary school | 1 854 229 (34) |
| | Missing | 109 072 (2) |
| Age group (years) | 20–29 | 1 090 723 (20) |
| | 30–39 | 1 254 332 (23) |
| | 40–49 | 1 254 332 (23) |
| | 50–59 | 1 199 796 (22) |
| | 60–64 | 708 970 (13) |
| Marital status | Married | 2 345 055 (43) |
| | Unmarried | 2 345 055 (43) |
| | Divorced | 654 434 (12) |
| | Widow/widower | 54 536 (1) |

also presented separately by men and women (see online supplemental files; online supplemental tables 1 and 2). Those aged 50–59 years had the highest prevalence in all registers except the Register of Convictions. The highest prevalence for both men and women was found in the Register of Medications (men 2.6%, women 1.2%), followed by the Inpatient and Outpatient Register and the Social Insurance Register. The lowest prevalence was found in the Register of Convictions. Comparing all registers combined with the register with the highest prevalence increased total prevalence by 60% for men and by 66% for women. Prevalence in the different age groups increased correspondingly from 24% to 67%, with the largest increase among people aged 30–39 years. Table 2 also shows the correlation between the registers—that is, the extent to which persons with an indication of alcohol use disorder in one register appear in other registers—and the unique contribution in percent of each register to a register indication. Register indication in the Inpatient and Outpatient Registers and the Register of Medications had the highest correlation; these three registers contributed the largest proportion of unique indications. The correlations between indications in other registers were strongest if a person appeared in the Register of Medications, and lowest if the person appeared in the Register of Convictions. The Social Insurance Register and Register of Convictions had the smallest unique contributions.

Individuals with an indication of alcohol use disorder were 167 175 and differed by age, sex, migrant status, education level and marital status (see table 3). Due

to this doubled prevalence among men, table 3 is also presented separately by men and women (see online supplemental file, online supplemental tables 3 and 4). Women with an indication of alcohol use disorder appeared more often in the Outpatient Register (8% more likely than men), but had a lower likelihood of being found in the Inpatient Register (4% less likely than men), Register of Medications (12% less likely than men) and Register of Convictions (74% less likely than men) (see table 3). Both for the total population with an indication of alcohol use disorders, and for men and women separately, the likelihood of having an indication in the Inpatient Register increased with age and the likelihood of an indication in the Register of Convictions decreased with age. Men with an indication of alcohol use disorder aged 20–29 or 40–49 years were more likely to be found in the Outpatient Register compared with those in other age groups, but this pattern was not seen in women. Both for the total population with an indication of alcohol use disorders, and for men and women separately, those aged 40–49 and 50–59 years appeared more often in the Register of Medications (table 3; see online supplemental file, online supplemental tables 3 and 4). Migrants with an indication of alcohol use disorder were more likely to appear in the Inpatient and Outpatient Registers, and they were also more likely to have an indication in the Register of Convictions, compared with those born in Sweden, with little difference between men and women. Persons born in Sweden with an indication of alcohol use disorder, however, were more likely to appear in the Register of Medications and in the Social Insurance Register compared with migrants (table 3). People with an indication of alcohol use disorder and those with postsecondary education appeared more often in the Register of Medications and the Outpatient Register compared with other education levels. However, among those with an indication of alcohol use disorder and presecondary education, it was more common to appear in the Register of Convictions and in the Inpatient Register. People with an indication of alcohol use disorder who were married were more likely to be found in the Register of Medications than individuals who were unmarried, divorced and widows and widowers (see table 3).

The prevalence of individuals with an indication of drug use disorder in any of the registers was 1.1% among women and 2.5% among men. Those aged 20–29 years had the highest prevalence in all registers, except in the Register of Medications (table 4). The prevalence of an indication of drug use disorder was consistently higher for men than women. Due to the more than doubled prevalence among men, table 4 is presented separately by men and women (see online supplemental file; online supplemental tables 5 and 6). For all registers, except in the Register of Medications, prevalence decreased with age. For men, the highest prevalence was found in the Register of Convictions (1.7%), followed by the Outpatient and Inpatient Registers and in the Register of Medications. For women, the highest prevalence was found in

**Table 2** Indication of alcohol use disorder in the total population.

| | Register | Inpatient | Outpatient | Medications | Social Insurance | Convictions | Any register |
|---|---|---|---|---|---|---|---|
| n (%) | | 66 940 (100) | 96 279 (100) | 103 199 (100) | 9 323 (100) | 1 837 (100) | 167 178 (100) |
| Sex | Men | 46 889 (1.69) | 65 445 (2.37) | 71 680 (2.59) | 6 415 (0.23) | 1 642 (0.06) | 114 789 (4.15) |
| | Women | 20 051 (0.75) | 30 834 (1.15) | 31 519 (1.17) | 2 908 (0.11) | 195 (0.01) | 52 389 (1.95) |
| Age (years) | 20–29 | 5 917 (0.91) | 10 282 (1.58) | 7 182 (1.1) | 385 (0.06) | 271 (0.04) | 16 847 (2.59) |
| | 30–39 | 9 053 (0.78) | 15 510 (1.33) | 15 780 (1.35) | 1 345 (0.12) | 377 (0.03) | 26 261 (2.25) |
| | 40–49 | 16 303 (1.27) | 24 430 (1.91) | 27 804 (2.17) | 3 055 (0.24) | 505 (0.04) | 41 229 (3.22) |
| | 50–59 | 19 326 (1.66) | 26 081 (2.25) | 2 9893 (2.57) | 3 319 (0.29) | 442 (0.04) | 45 448 (3.91) |
| | 60–64 | 16 341 (1.37) | 19 976 (1.67) | 22 540 (1.89) | 1 219 (0.1) | 242 (0.02) | 37 393 (3.13) |
| Migrant status | Born in Sweden | 57 316 (1.25) | 82 217 (1.8) | 91 418 (2) | 8 329 (0.18) | 1 527 (0.03) | 144 732 (3.16) |
| | Born abroad | 9 157 (1.16) | 13 256 (1.68) | 1 1222 (1.42) | 956 (0.12) | 288 (0.04) | 21 148 (2.68) |
| | Refugee | 467 (0.52) | 806 (0.89) | 559 (0.62) | 38 (0.04) | 22 (0.02) | 1 298 (1.44) |
| Education | Preupper secondary school | 20 819 (2.35) | 27 032 (3.05) | 26 908 (3.03) | 2 177 (0.25) | 756 (0.09) | 46 459 (5.23) |
| | Upper secondary | 3 5633 (1.36) | 50 424 (1.92) | 54 562 (2.08) | 5 466 (0.21) | 886 (0.03) | 88 073 (3.36) |
| | Postsecondary school | 9 720 (0.53) | 1 7802 (0.96) | 20 929 (1.13) | 1 649 (0.09) | 155 (0.01) | 30 936 (1.67) |
| | Missing | 768 (0.84) | 1 021 (1.11) | 800 (0.87) | 31 (0.03) | 40 (0.04) | 1 710 (1.87) |
| Marital status | Married | 14 687 (0.62) | 22 765 (0.96) | 30 818 (1.3) | 2 274 (0.1) | 312 (0.01) | 44 258 (1.87) |
| | Unmarried | 33 630 (1.42) | 49 598 (2.09) | 47 318 (2) | 4 525 (0.19) | 1 094 (0.05) | 82 454 (3.48) |
| | Divorced | 17 531 (2.69) | 22 561 (3.46) | 23 552 (3.61) | 2 417 (0.37) | 412 (0.06) | 37 987 (5.82) |
| | Widow(er) | 1 092 (1.55) | 1 355 (1.93) | 1 511 (2.15) | 107 (0.15) | 19 (0.03) | 2 479 (3.53) |
| Correlation coefficient | Inpatient | x | 0.54 | 0.41 | 0.25 | 0.04 | 0.63 |
| | Outpatient | | x | 0.49 | 0.22 | 0.04 | 0.75 |
| | Medications | | | x | 0.22 | 0.04 | 0.78 |
| | Social Insurance | | | | x | 0.02 | 0.23 |
| | Convictions | | | | | x | 0.10 |
| | Any registers | | | | | | x |
| Unique contribution to any register | | 15 066 (9.01) | 29 528 (17.66) | 45 358 (27.13) | 808 (0.48) | 196 (0.60) | x |

Number (n) and per cent (%) with register indication by age group, migrant status, education, marital status and phi (the correlation coefficient) between the registers, and the unique contribution of the specific register to all registers combined (any registers) in per cent (%) for the following registers: Inpatient, Outpatient, the Register of Medications (Medications), the Social Insurance Register (Social Insurance), the Register of Convictions (Convictions) and any register.

the Outpatient Register (0.6%), followed by the Inpatient Register, the Register of Convictions and in the Register of Medications. Both men and women had the lowest prevalence in the Social Insurance Register. For drug use disorder, the prevalence increased by 45% for men and by 80% for women if all registers were combined, instead of the register with the highest prevalence alone. The prevalence in different age groups increased correspondingly by between 30% and 109%, with the largest increase among people in the 50–59 years age group.

There was a correlation between having an indication of a drug use disorder between the registers. The correlation was highest between the Inpatient and Outpatient Registers (0.56). The correlation between having an indication in another register was strongest if the individual appeared in the Register of Convictions (0.76) and lowest

if they appeared in the Social Insurance Register (0.19). The register contributing the highest proportion of individual cases for men was the Register of Convictions, and Outpatient Care registers contributed the highest proportion of individual cases for women. The register contributing the smallest proportion of cases was the Register of Social Insurance, for both sexes.

Individuals who were found to have an indication of drug use disorder (n=99 529) differed by age, sex, migrant status, education level and marital status (see table 5). Due to the more than doubled prevalence among men, table 5 is also presented separately by men and women (see online supplemental file; online supplemental tables 7 and 8). In terms of sex differences among those with an indication of drug use disorder, women were more likely to appear in the Outpatient and Inpatient Registers as

Table 3 The association between socioeconomic factors and the likelyhood of being listed in a sepcific register among those with an indication of alcohol use disorder (n=167 175) by the following registers: Inpatient Care, the Outpatient care, the Register of Medications (Medications), the Social Insurance Register (Social Insurance) and the Register of Convictions (Convictions), presented with ORs with 95% CI, by age group, sex, migrant status, education level and marital status included in the multivariable models

| | | Inpatient OR (95% CI) | Outpatient OR (95% CI) | Medications OR (95% CI) | Social Insurance OR (95% CI) | Convictions OR (95% CI) |
|---|---|---|---|---|---|---|
| Age group (years) | 20–29 | 0.62 (0.60 to 0.65) | 1.07 (1.03 to 1.11) | 0.46 (0.45 to 0.48) | 0.30 (0.27 to 0.34) | 1.63 (1.38 to 1.93) |
| | 30–39 | 0.67 (0.65 to 0.69) | 1.02 (0.99 to 1.06) | 0.86 (0.83 to 0.89) | 0.70 (0.65 to 0.75) | 1.53 (1.33 to 1.78) |
| | 40–49 | 0.87 (0.85 to 0.90) | 1.07 (1.04 to 1.10) | 1.11 (1.08 to 1.15) | 1.01 (0.96 to 1.07) | 1.34 (1.17 to 1.52) |
| | 50–59 | 1 | 1 | 1 | 1 | 1 |
| | 60–64 | 1.09 (1.06 to 1.13) | 0.88 (0.86 to 0.91) | 0.74 (0.72 to 0.76) | 0.43 (0.40 to 0.46) | 0.64 (0.54 to 0.75) |
| Sex | Female | 0.96 (0.94 to 0.98) | 1.08 (1.06 to 1.10) | 0.88 (0.86 to 0.90) | 0.99 (0.94 to 1.03) | 0.26 (0.22 to 0.30) |
| | Male | 1 | 1 | 1 | 1 | 1 |
| Migrant status | Migrant | 1.15 (1.12 to 1.19) | 1.29 (1.25 to 1.33) | 0.63 (0.61 to 0.65) | 0.77 (0.72 to 0.83) | 1.27 (1.12 to 1.45) |
| | Refugee | 0.93 (0.83 to 1.04) | 1.25 (1.12 to 1.40) | 0.39 (0.35 to 0.44) | 0.46 (0.34 to 0.64) | 1.28 (0.84 to 1.97) |
| | Sweden | 1 | 1 | 1 | 1 | 1 |
| Education | Preupper secondary school | 1.17 (1.15 to 1.20) | 1.04 (1.01 to 1.06) | 0.90 (0.88 to 0.92) | 0.80 (0.76 to 0.84) | 1.58 (1.43 to 1.75) |
| | Upper secondary | 1 | 1 | 1 | 1 | 1 |
| | Postsecondary school | 0.69 (0.67 to 0.71) | 1.06 (1.03 to 1.09) | 1.23 (1.19 to 1.26) | 0.87 (0.82 to 0.92) | 0.59 (0.49 to 0.70) |
| | Missing | 1.30 (1.17 to 1.43) | 0.98 (0.88 to 1.08) | 0.81 (0.73 to 0.89) | 0.40 (0.28 to 0.58) | 1.82 (1.31 to 2.52) |
| Marital status | Single | 1.56 (1.52 to 1.60) | 1.39 (1.35 to 1.43) | 0.64 (0.62 to 0.66) | 1.11 (1.05 to 1.18) | 1.29 (1.13 to 1.48) |
| | Married | 1 | 1 | 1 | 1 | 1 |
| | Divorced | 1.62 (1.58 to 1.67) | 1.39 (1.35 to 1.42) | 0.73 (0.71 to 0.75) | 1.25 (1.18 to 1.32) | 1.56 (1.34 to 1.81) |
| | Widow/widower | 1.41 (1.29 to 1.53) | 1.18 (1.09 to 1.28) | 0.79 (0.73 to 0.86) | 1.06 (0.87 to 1.30) | 1.69 (1.06 to 2.71) |

well as in the Register of Medications compared with men, while men were more likely to appear in the Register of Convictions. Those aged 20–29 years with an indication of drug use disorder were more likely to appear in the Register of Convictions. Persons with an indication of drug use disorder aged 30–49 years were more likely than other groups to be found both in the Register of Convictions (together with those aged 20–29 years) and to have an indication in the Social Insurance Register. Individuals in the oldest age group (60–64 years) with an indication of drug use disorder were more likely to appear in the Register of Medications (see table 5). Male migrants with an indication of drug use disorder were more likely to appear in the Register of Convictions compared with individuals born in Sweden, but this was not seen among women (see online supplemental file; online supplemental tables 7 and 8). Those with a presecondary education were more likely to appear in all registers, except the Social Insurance Register, compared with those with a higher education. Unmarried and divorced individuals with an indication of drug use disorder were more likely to appear in the Inpatient Register and in the Register of Convictions compared with married individuals. Married individuals with an indication of drug use disorders appeared more often than other marital statuses in the Register of Medications.

## DISCUSSION
The study demonstrates that the proportion of individuals with an indication of alcohol or drug use disorder differs between registers and that there is a correlation between the registers. This means that those with a register indication of substance use in one register sometimes—but not always—appear in one or more other registers, thus the proportion increases when registers are added together. The proportion of the population with an indication of alcohol or drug use disorders increases by between 45% and 80% when compared with only the register showing the highest prevalence. For alcohol use disorder, the three healthcare registers (ie, the Inpatient and the Outpatient Registers and the Register of Medications) had the highest correlations; however, despite this correlation, the three healthcare registers also had the largest proportions of unique alcohol use disorder indications. The registers for Social Insurance and Convictions contribute less than 1% each. Still, if the Social Insurance Register is not included, this could bias the results towards missing indications of alcohol use disorders among individuals who are single or divorced. If the Register of Convictions is excluded, this could bias the results towards missing indications of alcohol use disorder among younger individuals (up to 49 years), those not born in Sweden, those without a higher education and those currently unmarried. For drug use

**Table 4** Indication of alcohol use disorder in the total population.

| | Register | Inpatient | Outpatient | Medications | Social Insurance | Convictions | Any registers |
|---|---|---|---|---|---|---|---|
| Total number | n (%) | 32 175 (100) | 46 123 (100) | 13 293 175 (100) | 3 559 (100) | 57 211 (100) | 99 552 (100) |
| Sex | Men | 20 365 (0.74) | 29 671 (1.07) | 8 117 (0.29) | 2 449 (0.09) | 48 155 (1.74) | 69 928 (2.53) |
| | Women | 11 810 (0.44) | 16 452 (0.61) | 5 176 (0.19) | 1 110 (0.04) | 9 056 (0.34) | 29 624 (1.1) |
| Age (years) | 20–29 | 7 879 (1.21) | 11 103 (1.71) | 1 354 (0.21) | 1 032 (0.16) | 20 788 (3.2) | 26 984 (4.15) |
| | 30–39 | 8 669 (0.74) | 12 571 (1.08) | 2 891 (0.25) | 1 215 (0.1) | 17 197 (1.47) | 26 026 (2.23) |
| | 40–49 | 7 766 (0.61) | 11 201 (0.88) | 3 154 (0.25) | 814 (0.06) | 11 527 (0.9) | 22 060 (1.73) |
| | 50–59 | 5 342 (0.46) | 7 811 (0.67) | 3 074 (0.26) | 422 (0.04) | 6 626 (0.57) | 16 279 (1.4) |
| | 60–64 | 2 519 (0.21) | 3 437 (0.29) | 2 820 (0.24) | 76 (0.01) | 1 073 (0.09) | 8 203 (0.69) |
| Migrant status | Born in Sweden | 26 897 (0.59) | 38 088 (0.83) | 11 084 (0.24) | 3 108 (0.07) | 44 336 (0.97) | 79 634 (1.74) |
| | Born abroad | 3 585 (0.58) | 6 866 (0.87) | 1 805 (0.23) | 396 (0.05) | 11 032 (1.4) | 17 209 (2.18) |
| | Refugee | 693 (0.77) | 1 169 (1.3) | 404 (0.45) | 55 (0.06) | 1 843 (2.05) | 2 709 (3.01) |
| Education | Preupper secondary school | 12 867 (1.45) | 17 702 (1.99) | 4 852 (0.55) | 874 (0.12) | 24 250 (2.73) | 36 557 (4.12) |
| | Upper secondary | 15 214 (0.58) | 22 070 (0.84) | 6 300 (0.24) | 1 943 (0.07) | 28 197 (1.07) | 50 019 (1.91) |
| | Postsecondary school | 3 475 (0.19) | 5 479 (0.3) | 1 989 (0.11) | 516 (0.03) | 3 343 (0.18) | 10 948 (0.59) |
| | Missing | 619 (0.68) | 872 (0.95) | 152 (0.17) | 26 (0.03) | 1 321 (1.55) | 2 028 (2.21) |
| Marital status | Married | 4 896 (0.21) | 7 159 (0.3) | 4 030 (0.17) | 513 (0.02) | 5 067 (0.21) | 16 050 (0.68) |
| | Unmarried | 21 510 (0.91) | 30 989 (1.31) | 6 603 (0.28) | 2 598 (0.11) | 10 196 (1.91) | 67 389 (2.85) |
| | Divorced | 5 508 (0.84) | 7 607 (1.17) | 2 433 (0.37) | 432 (0.07) | 6 762 (1.04) | 15 394 (2.36) |
| | Widow(er) | 261 (0.37) | 368 (0.52) | 227 (0.32) | 16 (0.02) | 165 (0.23) | 719 (1.02) |
| Correlation coefficient | Inpatient | x | 0.56 | 0.23 | 0.20 | 0.33 | 0.56 |
| | Outpatient | | x | 0.27 | 0.21 | 0.36 | 0.68 |
| | Medications | | | x | 0.10 | 0.15 | 0.36 |
| | Social Insurance | | | | x | 0.12 | 0.19 |
| | Convictions | | | | | x | 0.76 |
| | Any registers | | | | | | x |
| Unique contribution to any register | | 7 948 (7.99) | 15 915 (15.99) | 6 283 (6.31) | 506 (0.51) | 35 723 (35.88) | x |

Number (n) and per cent (%) with register indication by age group, migrant status, education, marital status and phi (the correlation coefficient) between registers and the unique contribution of each specific register to all registers combined (any registers) in per cent (%) for the following registers: Inpatient Care, Outpatient Care, the Register of Medications (Medications), the Social Insurance Register (Social Insurance), the Register of Convictions (Convictions) and any register.

disorder, the highest correlation with other registers was when an individual appeared in the Outpatient Register or the Register of Convictions. These two registers also contribute the most independent indications. The Social Insurance Register contributes less than 1%; however, if excluded, this could bias the results towards missing indications of drug use disorders among individuals younger than 49 years and those with a higher education.

### Findings in context

In this study, using all registers showed the proportion of the population with an indication of alcohol use disorder as 4.2% for men and 2.0% for women. This can be compared with a cross-sectional study performed by Andréasson *et al*[3] based on survey data from randomly selected 19–70 year-olds from 12 Swedish municipalities, where researchers found that the proportion classified as having an alcohol use disorder was 4.9% for men and 3.2% for women. The same study also found that an average of 4% of the population in Sweden meet at least three criteria for alcohol use disorder, while among 19–25 year-olds, the prevalence was higher (around 11%), and in this group, there was no statistical difference between women and men, a difference that was found in all other age groups.[3] These findings were also in line with the results from the 2017 survey 'Vanor och konsekvenser' (in English 'Habits and consequences').[6] The difference in prevalence between the two other studies and our study, especially in the case of women, indicates that an

**Table 5** The association between socioeconomic factors and the likelyhood of being listed in a sepcific register among those with an indication of drug use disorder (n=99 529) by the following registers: Inpatient Care, Outpatient Care, the Register of Medications (Medications), the Social Insurance Register (Social Insurance) and the Register of Convictions (Convictions) and any register (all registers combined) presented with ORs with 95% CI, including age group, sex, migrant status, education level and marital status in the multivariable models

| | | Inpatient | Outpatient | Medications | Social Insurance | Convictions |
|---|---|---|---|---|---|---|
| | | OR (95% CI) | OR (95% CI) | OR (95% CI) | OR (95% CI) | OR (95% CI) |
| Age group (years) | 20–29 | 0.86 (0.82 to 0.90) | 0.77 (0.74 to 0.80) | 0.26 (0.24 to 0.28) | 1.74 (1.53 to 1.97) | 3.82 (3.64 to 4.01) |
| | 30–39 | 1.07 (1.02 to 1.12) | 1.05 (1.01 to 1.09) | 0.57 (0.54 to 0.61) | 2.00 (1.78 to 2.26) | 2.57 (2.46 to 2.69) |
| | 40–49 | 1.13 (1.08 to 1.18) | 1.13 (1.09 to 1.18) | 0.73 (0.69 to 0.77) | 1.49 (1.32 to 1.68) | 1.61 (1.54 to 1.69) |
| | 50–59 | 1 | 1 | 1 | 1 | 1 |
| | 60–64 | 0.86 (0.81 to 0.91) | 0.74 (0.70 to 0.78) | 1.97 (1.85 to 2.09) | 0.33 (0.26 to 0.42) | 0.25 (0.23 to 0.27) |
| Sex | Female | 1.61 (1.56 to 1.66) | 1.69 (1.64 to 1.74) | 1.27 (1.22 to 1.32) | 1.07 (1.00 to 1.16) | 0.22 (0.22 to 0.23) |
| | Male | 1 | 1 | 1 | 1 | 1 |
| Migrant status | Born abroad | 0.72 (0.69 to 0.75) | 0.74 (0.71 to 0.76) | 0.73 (0.69 to 0.77) | 0.58 (0.52 to 0.64) | 1.38 (1.32 to 1.43) |
| | Refugee | 0.73 (0.67 to 0.80) | 0.90 (0.83 to 0.97) | 1.20 (1.08 to 1.35) | 0.50 (0.38 to 0.65) | 1.40 (1.28 to 1.54) |
| | Sweden | 1 | 1 | 1 | 1 | 1 |
| Education | Preupper secondary school | 1.30 (1.26 to 1.34) | 1.25 (1.22 to 1.29) | 1.21 (1.16 to 1.27) | 0.74 (0.69 to 0.80) | 1.44 (1.39 to 1.49) |
| | Upper secondary school | 1 | 1 | 1 | 1 | 1 |
| | Postsecondary school | 1.01 (0.96 to 1.05) | 1.20 (1.15 to 1.26) | 1.13 (1.06 to 1.20) | 1.36 (1.23 to 1.50) | 0.46 (0.44 to 0.49) |
| | Missing | 1.27 (1.15 to 1.41) | 1.23 (1.12 to 1.34) | 0.89 (0.75 to 1.05) | 0.36 (0.25 to 0.54) | 1.22 (1.09 to 1.36) |
| Marital status | Single | 1.14 (1.09 to 1.19) | 1.19 (1.14 to 1.24) | 0.56 (0.53 to 0.59) | 0.91 (0.82 to 1.01) | 2.15 (2.05 to 2.25) |
| | Married | 1 | 1 | 1 | 1 | 1 |
| | Divorced | 1.25 (1.19 to 1.31) | 1.20 (1.15 to 1.26) | 0.51 (0.48 to 0.54) | 0.95 (0.83 to 1.08) | 2.12 (2.01 to 2.23) |
| | Widow/widower | 1.17 (1.00 to 1.37) | 1.21 (1.04 to 1.41) | 0.87 (0.74 to 1.03) | 1.05 (0.63 to 1.74) | 1.72 (1.41 to 2.10) |

individual may estimate their alcohol consumption as high without seeking care. The study by Andréasson *et al* also shows that the youngest group (19–25 years) had the highest prevalence of alcohol use disorder (around 1%), while our study shows that the age group 50–59 years has the highest prevalence (3.9%). The registers (except the Register of Convictions) are largely based on people who have been using healthcare, so the difference between the previous survey-based studies and the present study may be due to lower care-seeking for substance use disorder among younger age groups compared with older age groups. It could also be due to the fact that the same alcohol intake could have different consequences in different age groups. This would mean that a younger person drinking a hazardous amount might perceive this as not harmful, whereas an older person drinking the same amount may experience hazardous consequences more directly and is therefore more likely to seek care. It was only in the register unrelated to care seeking, the Register of Convictions, where the youngest age group had a higher likelihood compared with the older age groups.

We found that 2.5% of men and 1.1% of women had an indication of drug use disorder. This can be compared with the results from Central Association for Alcohol and Drug Information survey,[6] where drug dependence according to the Diagnostic and Statistical Manual of Mental Disorders 5[24] amounted to 1.9% among men and 1.6% among women. The prevalence among men was therefore higher according to register data than in the survey-based study, while the opposite applied to women. In the present study, the Register of Convictions was central to an indication of drug use disorders, and it was more common for men than for women to occur in this register. Among men, the highest prevalence of an indication of drug use disorder was captured by the Register of Convictions (1.7%), followed by the Outpatient Register (1.1%). Among women, most were captured in the Outpatient and Inpatient Registers (0.6% and 0.4%, respectively), followed by the Register of Convictions (0.3%). The correlation was strongest if an individual appeared in the Register of Convictions, but this register also captured cases that were not in other registers. The vast coverage of this register was expected, as both drug possession and drug use are illegal in Sweden.

The prevalence of indications of alcohol and drug use disorders differs between registers. For example, drug use disorder was captured by the Register of Convictions to a stronger extent than alcohol use disorder, which might be explained by the fact that alcohol is legal while drugs are illegal. This legal difference not only affects the availability of these substances and thus their use, their

problematic use and the need for care, but also which register identifies said use. In the case of alcohol use disorder, the Register of Medications had the highest prevalence among both men and women. In addition, the Register of Medications showed a different pattern than the Outpatient and Inpatient Care Registers. Among those with an indication of alcohol use disorder who were married, those born in Sweden and those with a postsecondary education had a higher likelihood to be found in this register but were less likely to be found in the Outpatient and Inpatient Registers. The finding that medications targeting alcohol use disorder are prescribed to a stronger extent among people born in Sweden and among people with higher education, compared with migrants and those with a shorter education, is in line with previous studies.[25]

## Strengths and limitations

This study examines the proportion of individuals with an indication of alcohol or drug use disorder during a specific period (period prevalence) without concern of previous or repeated use of healthcare or the social insurance system. Individuals found in multiple registers are likely to have more severe alcohol and drug use disorders. Although adding all the registers together increases the prevalence of indications of substance use disorders, this study probably underestimates the prevalence of early stages of substance use disorder, as those at this stage possibly are less likely to seek care and might be more careful in terms of, for instance, drunk driving. Also, we have not included diagnoses from primary care, therefore, some individuals seeking care at an early stage may not be detected. However, as we include the register of Medications where all prescribed medications are registered, the suggested the underestimation of patients with substance use disorder in primary care is limited to patients who have not been prescribed medication, nor been referred to outpatient care. Another limitation is the age span of 20–64 years as both alcohol and drug use disorders are common among adolescents and young adults and among the elderly.

A valid substance use diagnosis can be described as a diagnosis that meets the ICD-10 criteria. The validity of indications of alcohol and drug use disorder in this study depends on the validity of the different registers. The diagnoses in the Inpatient and Outpatient Registers were made by physicians and can therefore be considered valid.[13] Being offered sickness or activity compensation by the Swedish Social Insurance Agency is preceded by a process in which physicians are involved, and although there are some differences in insurance medical practice over the country, these indications can also be considered valid. With regard to the Register of Medications, none of the medications used to treat alcohol use disorder included in the study have other uses according to the Swedish Pharmacopeia Information guide,[26] and all were prescribed by physicians and can therefore be considered valid. Some of the medications included as a marker for drug use disorder in the study can, in some special cases, be used for pain relief, which reduces the validity. With regard to the Register of Convictions, this differs from the other registers in that it only measures one aspect of substance use disorder, namely the ICD-10 criterion of continued drinking or use despite negative effects, and is therefore only a proxy variable, that is, a variable that replaces another variable when it cannot be directly observed. It can, accordingly, be considered to have the lowest validity. Only cases when a person had more than two convictions for relevant alcohol-related or drug-related crime during a calendar year were counted as an indication of substance use disorder. More individuals with alcohol or drug use disorder would have been identified from the Register of Convictions if the threshold of number of convictions during a calendar had been lower; however, this practice could also have increased the number of false positives and was not graded according to the ICD-10 criteria of substance use disorder.[23] Having three convictions for severe forms of alcohol-related crime is relatively rare, while for aggravated drunk driving, there is a suspension period—that is, a time when the driving licence is revoked—that typically amounts to 12 months. Regarding the indication of drug use disorder in the Register of Convictions, it is not certain that possession or sale of drugs involves either alcohol use or drug use disorder, although studies show that people who sell drugs usually also use drugs.[27 28] There are other variables that could possibly have also been included as an indication of substance use disorder, such as intoxication (from the Inpatient or Outpatient Registers) where more than two occurrences could be an indicator of possible SUD. With regard to education, this variable is more often missing among migrants compared with individuals born in Sweden, and this could bias the results when it comes to education.

## CONCLUSION

The conclusion of this study is that the use of a combination of relevant population-based registers improves the ability to describe the prevalence of alcohol and drug use disorder in the population. Another effect of combining multiple registers is that indications of alcohol or drug use disorder among younger people and migrants, as well as among married individuals and those with a higher education, increased in comparison with only using one register. The registers contribute differently to indications of alcohol and drug use disorder. For instance, the Register of Convictions contributes less than 1% of the unique indications of alcohol use disorder but contributes more than 35% of the unique indications of drug use disorders. The study increases knowledge of how the use of different administrative registers creates a more accurate picture of the prevalence and differences in substance use disorder between different demographic groups in the population, although it is important to keep in mind the validity of the different registers.

**Acknowledgements** Parts of the results of this study have been published with other research questions in a report published by the Public Health Agency of Sweden in Swedish (https://www.folkhalsomyndigheten.se/publicerat-material/publikationsarkiv/a/alkohol-ochnarkotikaberoende/).

**Contributors** AL, A-CH and CD conceived the study. AL and A-CH designed the study. A-CH obtained funding from FORTE and the Public Health Agency of Sweden. CD acquired the cohort data. AL prepared the data. AL conducted the statistical analyses and drafted the data tables. AL, A-CH and A-KD interpreted the statistical analyses. AL and A-CH wrote the manuscript. All authors critically revised the paper for important intellectual content and approved the final version. ACH is the guarantor.

**Funding** A-CH is supported by the Swedish Council for Health, Working Life, and Welfare (FORTE, by Swedish acronym) (grant numbers 2016-00870). CD is supported by the Swedish Research Council (grant number 5232010-1052). This work was partially funded by the Public Health Agency of Sweden.

**Competing interests** None declared.

**Patient and public involvement** Patients and/or the public were not involved in the design, or conduct, or reporting, or dissemination plans of this research.

**Patient consent for publication** Not applicable.

**Ethics approval** The project and the register linking were approved by the Regional Ethical Review Board in Stockholm (dnr. 2016/987-31).

**Provenance and peer review** Not commissioned; externally peer reviewed.

**Data availability statement** Data are available on reasonable request. Data is available on request for any interested researchers to allow replication of results through the Swedish National Data Service, provided all ethical and legal requirements are met. Detailed information on data application can be found at https://www.registerforskning.se/en/.

**ORCID iDs**
Andreas Lundin http://orcid.org/0000-0002-2318-2113
Anna-Karin Danielsson http://orcid.org/0000-0003-4932-4607
Anna-Clara Hollander http://orcid.org/0000-0002-1246-5804

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
