## [Reviewer comments · BMJ Open]

ARTICLE DETAILS

TITLE (PROVISIONAL)	Indications of alcohol- or drug use disorders in five different national registers in Sweden—a cross sectional population-based study.
AUTHORS	Hollander, Anna-Clara; Lundin, Andreas; Danielsson, Anna-Karin; Dalman, Christina

VERSION 1 – REVIEW

REVIEWER	Bogstrand, Stig Tore Oslo Universitetssykehus, Forensic Sciences
REVIEW RETURNED	26-Jan-2023

GENERAL COMMENTS	This is a well written manuscript with an interesting method of estimating the correlation of alcohol or drug use disorders between different registers. Here are some suggestions for improvement: Title The title is long. Suggest shortening it down a bit. Abstract Strengths and limitations of this study – page 4 Bullet point 2-3, are more descriptive than related to strengths and limitation Mention the high quality registers as a strength Major Background Elaborate on the point made in line 12-13 (p.4) starting with Substance use... it is a bit unclear as of now. Selection bias is one of the major limitations to surveys in this topic, should be addressed in the second paragraph (p.4) Methods Some bullet points seem to be missing in line 7-9 (p.5) An explanation of which data you get from Försäkringskassan would be beneficial. Is it unemployment rate? Other social services? Which variables were included? In both alcohol and drug use disorders, described on page 6 poisoning diagnosis should be mentioned. For alcohol use it might be less relevant, but for drug use disorders it is an important health outcome closely related to drug use disorders. It is interesting to understand how the authors have assessed it. More than one poisoning event, with hospital treatment would be a strong indication of AUD/DUD In relation to this in line 44-50 p.5 please indicate which ICD 10 diagnosis are referred to (F1X.X) and why are the intoxication codes excluded (T4, X41, X42, X44, etc.)? They indicate drug use disorder as well.
---

	Page 5 line 38: why persons between 20-64, I cannot see this explained anywhere? Page 6 line 45: your final age category is 60-65 while you describe including only patients aged 60-64. Why did you only include more than two convictions due to drugs during one calendar year? That period could have been larger, it is a bit strict. People with several convictions over several years could also be said to have drug use disorders. Results Table 1 could have two current digits; it would be easier to read without decimals. Explanation to table 2 and 4 is unclear, is the phi coefficient also presented in the table? Use N/A or X or another indication instead of 1. "Social Insurance" and "Sick leave" probably refers to the same register. One of the terms should be selected. The explanation of table 2 is unclear is the phi coefficient also presented in the table? Discussion First two paragraphs must be shortened to one and much more concise on the main findings, not a repetition of all findings Limits p.14-15 – you should mention that people at an early stage of drug use disorders are excluded in this study. A primary care register seems to be lacking, how does that affect the study? The sub-section can be shorter Minor Page 2 Methods line 15: only 4 registers seem to be noted of the 5. Page 2 Methods line 19: superfluous "a". Page 2 Results line 26-28: both space and no space in different occurrences of number and %. Page 2 Conclusions line 35: Is rehabilitation relevant to the findings? Page 4 DALYs abbreviation in line 6 Page 4 GP in line 3 Page 5 Parenthesis in line 25 Missing explanation to the asterisk in table 1 Page 8 Indication-s line 41 Table 3 and 5 use bold typing on the significant OR's or an other indication. Define: Page 7 line 17: Title of table 1: I presume you mean "characterization" rather than "characterise". Page 8: "...with an indication..." rather than "...an indications.." Page 10 line 18: "There" rather than "The", but I still a bit unclear. Page 10 line 52: imprecise sentence ("With"/"who"). Page 10 line 59: "sex" or "gender"? ("gender" is used in the abstract). Page 14 line 3: maybe translate the study name to English, unless this is done elsewhere. Page 14 line 4: what is meant by individuals not seeking "compensation"? Page 14 line 10: "based on people who have sought care"? Page 14 Great correlation, use strong correlation line 33 Page 14 line 48: "while these groups were less likely to..."? Page 14 line 49: "medications" rather than "medication".
--	---

	Page 15 line 24: “during a calendar year...”? Page 16 line 19: “FORTE” previously described?
--	---

REVIEWER	Lönn, Sara Center for Primary Health Care Research, Lund University
REVIEW RETURNED	22-Feb-2023

GENERAL COMMENTS	The Swedish registers is an important data source in part of the epidemiological research. An understanding of what different registrations in these registers represent is therefore important. However, for this paper to be suitable as a scientific publication I would expect a more profound analysis and discussion. It is already known that combining several registers increase the registrations of AUD as well as DUD. Nevertheless, it is of value to further describe how registrations in different registers correlate, to what extent they overlap and how this influence the interpretation of results from Swedish register-based studies. With more knowledge on this topic one could understand whether all registers are needed or if some can be left out. Or, if one does not have access to all registers, how big of a problem is this? Would it bias your estimates? However, these questions are not addressed well enough in the manuscript. Below I have tried to describe in more detail a few of the points that I found problematic in the manuscript. The descriptive statistics is not detailed enough. The number of cases need to be presented in Table 2 and 4 and the number of cases and proportions should to be presented for all groups, not only sex and age. Further, you need to present the proportion of unique registrations for each of the five registers. This is to understand the possible underestimation by group if, for example you do not have access to the social insurance or conviction registers. The overlap between registers needs to be described in much more detail. The author state that “All regression models were adjusted for age, sex, migrant status, education level and marital status.” Do you mean that the models are full models with all the variables included? This is not an “adjusted model” and should therefore not be referred to as such. If this is the correct interpretation, then adding univariable analysis might be informative. For example, education is rarely missing for Swedish born subject why having level of education in a model when you want to understand then influence of immigration might be problematic. Considering the gender differences that has been seen for these two outcomes, the tables and analysis need to be stratified by gender. What is the rational for not including adolescence, age 15 to 20, and people over 65? I understand these individuals are not found in the social insurance register, but DUD is quite common under the age of 20 and AUD over age 65. You need to mention which years the different registers cover and how does this effect your conclusions.
---

	Minor comment Please avoid terms like 'determine', as it is a bit too strong in this context. 'Investigating' would be preferable.
--	---

VERSION 1 – AUTHOR RESPONSE

Reviewer 1

- This is a well written manuscript with an interesting method of estimating the correlation of alcohol or drug use disorders between different registers.

We thank reviewer 1 for these kind words.

- The title is long. Suggest shortening it down a bit.

We thank the reviewer 1 for this suggestion: "Indications of alcohol- or drug use disorders in five different national registers in Sweden - a cross sectional population-based study." See title page.

- Bullet point 2-3, are more descriptive than related to strengths and limitation

We agree with reviewer 1 and have updated this section:

"Strengths and limitations of this study

- *The study has a large sample size as it included all individuals living in Sweden aged 20–64 in 2006 (n = 5 453 616).*
- *The five national registers included in the study are of high quality.*
- *A limitation of the study is that it examines the period prevalence of substance use disorder and does not take previous use into account.*

The validity of indications of alcohol- and drug use disorder depends on the validity of the different registers."

- Elaborate on the point made in line 12-13 (p.4) starting with Substance use... it is a bit unclear as of now.

We agree with reviewer 1 and due to this and other comments with regards to the first paragraph we have now rewritten the sentence to: "The term substance use disorder refers to both substance abuse and substance dependence and although studies of it in a population commonly use surveys, registry-based studies provide an additional way of studying substance use disorders [6-10]."

- Selection bias is one of the major limitations to surveys in this topic, should be addressed in the second paragraph (p.4)

We agree with reviewer 1 and we have added this the second paragraph: "However, disadvantages include selection bias, as some socioeconomic groups are more less likely to respond to surveys than others, the decreasing response rate in population surveys in general [11], and that heavy substance use can affect memory [12], increasing the risk of recall bias."

- Some bullet points seem to be missing in line 7-9 (p.5)

We thank reviewer 1 for pointing this out and have now updated this part of the Methods section without bullet points all together.

- An explanation of which data you get from Försäkringskassan would be beneficial. Is it unemployment rate? Other social services? Which variables were included?

We agree with reviewer 1 and have now rewritten this section with a richer description: "We also included the Swedish Social Insurance Agency's (Försäkringskassan) [26] registers of sickness or activity compensation included in Mikrodata for analysis of social insurance (henceforth called the Social Insurance register) were we used the variables Sickness absence and Disability pension. Sickness absence refers to the temporarily reduced work capacity due to alcohol and drug use disorders. Disability pension is a public financial support provided to the individuals who permanently leave the labour market before the age of statutory retirement usually preceded by sick leave."

- In both alcohol and drug use disorders, described on page 6 poisoning diagnosis should be mentioned. For alcohol use it might be less relevant, but for drug use disorders it is an important health outcome closely related to drug use disorders. It is interesting to understand how the authors

have assessed it. More than one poisoning event, with hospital treatment would be a strong indication of AUD/DUD

We agree with reviewer 1 that poisoning could be strong indicator of especially DUD, however we are not so sure in terms of AUD as this could also be an indicator of being novel to the use of alcohol and thus, we did not include it among the diagnoses. However, we consider this a limitation and have added a line about this in under limitations "The Register of Convictions were used as proxy variable and there are other proxy variables that could have been included too such as intoxication (from the Inpatient or Outpatient register) were more than two times could be an indicator of possible SUD."

- In relation to this in line 44-50 p.5 please indicate which ICD 10 diagnosis are referred to (F1X.X) and why are the intoxication codes excluded (T4, X41, X42, X44, etc.)? They indicate drug use disorder as well.

We thank reviewer 1 for this clarifying suggestion and this is now specified at page 5 "In the Patient registers and the Social Insurance register, an indication of drug use disorder was an ICD-10 diagnosis of chapters F11 – F14, F16, F18 – F19, which are Mental disorders and behavioural disorders caused by:

- *opioids (F11)*
- *cannabinoids (F12)*
- *sedatives or hypnotics (F13)*
- *cocaine (F14)*
- *hallucinogens (F16)*
- *volatile solvents (F18)*
- *psychoactive substances (F19)".*

With regards to the question about intoxication, see previous question.

- Page 5 line 38: why persons between 20-64, I cannot see this explained anywhere?
We appreciate this chance to clarify the manuscript and, in the Methods section, under the heading Population, we have now added: ". The lower age limit was determined by not having access to data from the child and adolescents' psychiatry from all over Sweden, and the higher as persons over 64 cannot be included in the register of Social Insurance due to the Swedish age of retirement in 2006. The population were then followed up in various registers regarding the indication of alcohol and drug use disorder between 2006 and 2016."

- Page 6 line 45: your final age category is 60-65 while you describe including only patients aged 60-64.

We thank reviewer 1 for pointing this out and have changed to 64 throughout.

- Why did you only include more than two convictions due to drugs during one calendar year? That period could have been larger, it is a bit strict. People with several convictions over several years could also be said to have drug use disorders.

We agree with reviewer 1 that it is a strict measure, however, the relational behind choosing this period is that we wanted it to be in line with the ICD-10 criteria for dependence "Continued intake despite adverse effects", it was only counted as a register indication if an individual had more than two convictions during the same calendar year. However, this is discussed under limitations "Only when a person had more than two convictions for relevant alcohol or drug related crime during the same calendar year was this counted as an indication of substance use disorder. It is possible that more individuals with alcohol or drug use disorder would have been identified from the register of convictions if the threshold of number of convictions during a calendar had been lower, however this practice could also have increased the false positives and not according to the ICD-10 criteria of substance use disorder [33]."

- Table 1 could have two current digits; it would be easier to read without decimals.

We agree and have updated the manuscript.

- Explanation to table 2 and 4 is unclear, is the phi coefficient also presented in the table?

We apologize for explanation, the phi coefficient is presented in the table 2 and 4, we have updated it now to be clearer and the explanation now reads: "Table 2/4. Indication of alcohol/drug use disorder in the total population. Total number n with indication, percent % with register indication by age-group, migrant status, education, marital status, phi - the correlation coefficient between the registers and the unique contribution of the specific register to all registers combined (Any registers) for the following registers: Inpatient care, Outpatient care, the Register of Medications (Medications), the Social Insurance register (Social Insurance), the Register of Convictions (Convictions), and Any register."

- Use N/A or X or another indication instead of 1.

We agree and have updated the manuscript.

- "Social Insurance" and "Sick leave" probably refers to the same register. One of the terms should be selected.

We agree and have updated the manuscript with Social Insurance throughout.

- The explanation of table 2 is unclear is the phi coefficient also presented in the table?

We apologize for explanation, the phi coefficient is presented in table 2, we have updated it now to be clearer and the explanation now reads: "Table 2/4. Indication of alcohol/drug use disorder in the total population. Total number n with indication, percent % with register indication by age-group, migrant status, education, marital status, phi - the correlation coefficient between the registers and the unique contribution of the specific register to all registers combined (Any registers) for the following registers: Inpatient care, Outpatient care, the Register of Medications (Medications), the Social Insurance register (Social Insurance), the Register of Convictions (Convictions), and Any register."

- First two paragraphs must be shortened to one and much more concise on the main findings, not a repetition of all findings

We thank reviewer 1 for this opportunity to improve this manuscript and have edited the first two paragraphs now reading "Alcohol and drug use cause a substantial disease burden globally, the composition and extent of which varies between countries [1]. Globally, in 2016, 4.2 % of all disability-adjusted life year (DALYs) were attributable to alcohol use, and 1.3 % to drug use [1] and the lost DALYs were due to the risk of injury, premature death, and other negative consequences [2]. However, the risk posed by using alcohol and drugs differs between men and women, by socioeconomic conditions, and for migrants by country of birth [3-5]. The term substance use disorder refers to both substance abuse and substance dependence and although studies of it in a population commonly use surveys, registry-based studies provide an additional way of studying substance use disorders [6-10]. There are many advantages to using survey data in studies of alcohol and drug use disorders in a population, such as the ability to study different degrees of the disorder as the participants describe their consumption and their related problems themselves. However, disadvantages of surveys include: decreasing response rate in population surveys in general [11], selection bias, as some socioeconomic groups are less likely to respond to surveys than others, and that heavy substance use can affect memory [12], increasing the risk of recall bias.

In Scandinavia and Finland, health care visits are recorded in local and national administrative registers covering the entire population. Such population-based registers have benefited psychiatric research immensely [13]. These health registers are generally considered valid [19], but using registers collected for administrative purposes in research can give misleading results if there is a systematic underutilisation of care in specific sub-groups. In Sweden to date, studies of substance use disorder have used the National Patient Register [6, 10, 20] from the perspective that the most severe cases of substance use disorder will require care. This register has been an important source of knowledge about the social determinants and consequences of severe alcohol- and drug use disorder. However, it has long been recognised that, in addition to detecting the most severe cases only, these registers possibly underestimate severe cases in specific sub-groups where under-utilisation is pronounced [21]. As register-based research could be a powerful tool for population-based studies of substance use disorders, there have been attempts to expand the use of population-registers by including additional registers with possibly less severe cases, such as registers with diagnoses from General practitioners (GP) surgeries [22], the Crime Prevention Council's register [8],

and the Register of prescribed and purchased medicine [9]. Studies using the additional registers recognise that appearing in the additional registers is not the same as being diagnosed with a substance use disorder but acknowledge that it is a possible indication of a substance use disorder. However, despite being used in studies of the determinants and consequences of a substance use disorder, to date this practice has not been tested empirically.”

- Limits p.14-15 – you should mention that people at an early stage of drug use disorders are excluded in this study.

We agree with reviewer 1 that this is missing and at page 16 we now write: “Although adding all the registers together increase the SUB prevalence, this study probably underestimates the prevalence at an early stage of substance use disorders as this group seek less care and might be more careful in terms of, for instance, drunk driving. Also, we have not included diagnoses from primary care, thus some individuals seeking care at an early stage of substance use disorder may be excluded in this study. However, as we include the register of Medications where all prescribed medication are registered, including medication prescribed in primary care (the first hand choice of treatment for alcohol use disorder in Swedish primary care [33]), this is limited to patients have not been purchasing prescribed medication nor been referred to outpatient care.”

- A primary care register seems to be lacking, how does that affect the study?

We thank reviewer 1 for this opportunity to improve this manuscript and have edited the first paragraphs under limitations now reading “Also, we have not included diagnoses from primary care, thus some individuals seeking care at an early stage of substance use disorder may be excluded in this study. However, as we include the register of Medications where all prescribed medication are registered, including medication prescribed in primary care (the first hand choice of treatment for alcohol use disorder in Swedish primary care [33]), this is limited to patients have not been purchasing prescribed medication nor been referred to outpatient care.” at page 16.

- Page 2 Methods line 15: only 4 registers seem to be noted of the 5.

We thank reviewer 1 for the chance to clarify this. The Patient register accurately contains two registers; one for inpatient care and another for specialized outpatient care thus its five in the list, this is now clearer in the text.

- Page 2 Methods line 19: superfluous “a”.

We agree and have updated the manuscript.

- Page 2 Results line 26-28: both space and no space in different occurrences of number and %.

We agree and have updated the manuscript.

- Page 2 Conclusions line 35: Is rehabilitation relevant to the findings?

We thank reviewer 1 for this interesting question but although the ultimate aim for research on substance use disorders should be rehabilitation, there sadly is not room for a discussion of this in this article.

- Page 4 DALYs abbreviation in line 6

We thank reviewer 1 for pointing this out and have now updated the manuscript.

- Page 4 GP in line 3

We thank reviewer 1 for pointing this out and have now updated the manuscript.

- Page 5 Parenthesis in line 25

We thank reviewer 1 for pointing this out and have now updated the manuscript.

- Missing explanation to the asterisk in table 1

We thank reviewer 1 for pointing this out and have now deleted the asterix from the manuscript, as the definition of refugee is already included in the Methods.

- Page 8 Indication-s line 41

We thank reviewer 1 for pointing this out and have now updated the manuscript.

- Table 3 and 5 use bold typing on the significant OR's or an other indication.

We thank reviewer 1 for pointing this out and have now updated the manuscript by not using bold in the table.

- Page 7 line 17: Title of table 1: I presume you mean “characterization” rather than “characterise”.
We thank reviewer 1 for pointing this out and have now updated the manuscript.
- Page 8: “...with an indication...” rather than “..an indications..”
We thank reviewer 1 for pointing this out and have now updated the manuscript.
- Page 10 line 18: “There” rather than “The”, but I still a bit unclear.
We thank reviewer 1 for pointing this out and have now updated the manuscript.
- Page 10 line 52: imprecise sentence (“With”/“who”).
We thank reviewer 1 for pointing this out and have now updated the manuscript.
- Page 10 line 59: “sex” or “gender”? (“gender” is used in the abstract).
We thank reviewer 1 for pointing this out and have now updated the manuscript with sex instead of gender.
- Page 14 line 3: maybe translate the study name to English, unless this is done elsewhere.
We thank reviewer 1 for pointing this out and have now translated it in the manuscript.
- Page 14 line 4: what is meant by individuals not seeking “compensation”?
We agree that this was unclear and thus deleted the word “compensation”
- Page 14 line 10: “based on people who have sought care”?
We agree that this was unclear and have updated the manuscript by changing “sought care” to “utilizing health care.”
- Page 14 Great correlation, use strong correlation line 33
We thank reviewer 1 for pointing this out and have now translated it in the manuscript.
- Page 14 line 48: “while these groups were less likely to...”?
We thank reviewer 1 for pointing this out and have now translated it in the manuscript.
- Page 14 line 49: “medications” rather than “medication”.
We thank reviewer 1 for pointing this out and have now updated the manuscript.
- Page 15 line 24: “during a calendar year...”?
We thank reviewer 1 for pointing this out and have now updated the manuscript.
- Page 16 line 19: “FORTE” previously described?
We thank reviewer 1 for pointing this out and have now updated the manuscript.

Reviewer 2

- It is already known that combining several registers increase the registrations of AUD as well as DUD. Nevertheless, it is of value to further describe how registrations in different registers correlate, to what extent they overlap and how this influence the interpretation of results from Swedish register-based studies. With more knowledge on this topic one could understand whether all registers are needed or if some can be left out. Or, if one does not have access to all registers, how big of a problem is this? Would it bias your estimates? However, these questions are not addressed well enough in the manuscript.

We thank reviewer 2 for these relevant questions and answering them have improved the manuscript significantly. To address them we have added a sentence in the Abstract, updated table 2 and 4 and edited the paragraph referring to table 2 and 4 in the Results section, and in the Discussion section we have edited both the first paragraph and the Conclusions.

In the Abstract we write under the Results section: “. The registers contributed to indications of alcohol and drugs use disorder differently.”.

With regards to Results we have added a section to table 2 and table 4 where we estimate the unique contribution of each specific register to the indications of alcohol/drug use disorders when taken all together. At page 14 in the Discussion part we write “For alcohol use disorder the three health care registers i.e., the Inpatient and the Outpatient registers and the Register of medications had the highest correlations, however, despite this correlation the three health care registers also had the largest proportions of unique alcohol use disorder indications. The registers for Social Insurance and

Convictions only contribute with less than one percent each. Still, if not including the Social Insurance register this could bias the results towards missing indications of alcohol use disorders among individuals who are single or divorced. If excluding the Register of Convictions this could bias the results towards missing indication of alcohol use disorder among the younger (up to 49), those not born in Sweden, those without a higher education and not currently married. For drug use disorder the highest correlation with other registers was when an individual appeared in the Outpatient register or the Register of Convictions. These two registers also contribute with most independent indications. The Social Insurance register contribute with less than one percent, however, if not including it among the registers this could bias the results towards missing indications of drug use disorders among individual younger than 49 years and with a higher education. ”

- The descriptive statistics is not detailed enough. The number of cases need to be presented in Table 2 and 4 and the number of cases and proportions should to be presented for all groups, not only sex and age.

We thank reviewer 2 for pointing this out and have now included percentage for all variable in table 2 and 4 in the manuscript and, for reasons of readability and space, we have included number of cases for each variables in table 2a and 2b and table 4a and 4b presented in the Supplementary files.

- Further, you need to present the proportion of unique registrations for each of the five registers. This is to understand the possible underestimation by group if, for example you do not have access to the social insurance or conviction registers.

We thank reviewer 2 for pointing this out and have now included this in table 2 and 4.

- The overlap between registers needs to be described in much more detail.

We agree with reviewer 2 that the correlation i.e. the extent to which persons with an indication of alcohol use disorder in one register appear in other registers needs to be discussed in more detail and thus we have added a section in the Discussion section and under Conclusions at page 14 “For alcohol use disorder the three health care registers i.e., the Inpatient and the Outpatient registers and the Register of medications had the highest correlations, however, despite this correlation the three health care registers also had the largest proportions of unique alcohol use disorder indications. The registers for Social Insurance and Convictions only contribute with less than one percent each. Still, if not including the Social Insurance register this could bias the results towards missing indications of alcohol use disorders among individuals who are single or divorced. If excluding the Register of Convictions this could bias the results towards missing indication of alcohol use disorder among the younger (up to 49), those not born in Sweden, those without a higher education and not currently married. For drug use disorder the highest correlation with other registers was when an individual appeared in the Outpatient register or the Register of Convictions. These two registers also contribute with most independent indications. The Social Insurance register contribute with less than one percent, however, if not including it among the registers this could bias the results towards missing indications of drug use disorders among individual younger than 49 years and with a higher education.”.

- The author state that “All regression models were adjusted for age, sex, migrant status, education level and marital status.” Do you mean that the models are full models with all the variables included? This is not an “adjusted model” and should therefore not be referred to as such. If this is the correct interpretation, then adding univariable analysis might be informative.

We agree with reviewer 2 that this is not adjusted models and have updated the text throughout. Unfortunately, there is not space to including a univariate analysis.

- For example, education is rarely missing for Swedish born subject why having level of education in a model when you want to understand then influence of immigration might be problematic. ‘

We agree that education could be problematic in register-based studies of migrants in Sweden, however Statistics Sweden converted education completed outside Sweden into equivalent levels of schooling in Sweden,

see <https://www.scb.se/contentassets/f0bc88c852364b6ea5c1654a0cc90234/dokumentation-av-lisa.pdf> page 44.

- Considering the gender differences that has been seen for these two outcomes, the tables and analysis need to be stratified by gender.

We agree with reviewer 2 and have added gender stratified tables for all tables in the manuscript (except table 1) in in the Supplementary files.

- What is the rational for not including adolescence, age 15 to 20, and people over 65? I understand these individuals are not found in the social insurance register, but DUD is quite common under the age of 20 and AUD over age 65.

We appreciate this chance to clarify the manuscript and, in the Methods section, under the heading Population, we have now added: "The lower age limit was determined by not having access to data from the child and adolescents' psychiatry from all over Sweden, and the higher as persons over 64 cannot be included in the register of Social Insurance due to the Swedish age of retirement in 2006. The population were then followed up in various registers regarding the indication of alcohol and drug use disorder between 2006 and 2016." We have also added under limitations: "Yet another limitation is the age span 20-64 as both alcohol and drug use disorders are common among adolescents and young adults and among elderly. "

- You need to mention which years the different registers cover and how does this effect your conclusions.

We thank reviewer two for a chance to clarify this, and we have now added the years when the registers started in the Methods section under Data sources. All registers were ongoing from start to end of study.

- Please avoid terms like 'determine', as it is a bit too strong in this context. 'Investigating' would be preferable.

We thank reviewer 2 for pointing this out and have now updated the manuscript.

VERSION 2 – REVIEW

REVIEWER	Lönn, Sara Center for Primary Health Care Research, Lund University
REVIEW RETURNED	07-Jul-2023

GENERAL COMMENTS	The authors have addressed most of my comments, or justified why they were not possible address. However, there is one important comment which I think was misunderstood and needs to be addressed. The number and proportion of unique cases should be reported for each cell as well, preferably among the Supplementary files as the other added tables. With regards to education, I understand that it might be converted into equivalent levels of schooling in Sweden but that is only if there is information on education to begin with. If you check in LISA you will find that education is more often missing for people born outside Sweden and by including this variables in an analysis you exclude a proportion of the immigrants, which could bias the results. In addition, a minor but important mistake that appeared after the revision. The terminology "multivariate model", is not correct in this situation (although I am aware that it is sometimes used) Please use the term multivariable instead (10.2105/AJPH.2012.300897).
---

VERSION 2 – AUTHOR RESPONSE

Reviewer Report

Reviewer: 1

The number and proportion of unique cases should be reported for each cell as well, preferably among the Supplementary files as the other added tables.

We agree with reviewer 1 that including unique cases is improving the manuscript and these numbers are now included in the manuscript see table 1, 2 and 4.

With regards to education, I understand that it might be converted into equivalent levels of schooling in Sweden but that is only if there is information on education to begin with. If you check in LISA you will find that education is more often missing for people born outside Sweden and by including this variables in an analysis you exclude a proportion of the immigrants, which could bias the results.

We agree with reviewer 1 and have added the sentence: "With regard to education, this variable is more often missing among migrants compared to individuals born in Sweden, and this could bias the results when it comes to education." at page 18.

In addition, a minor but important mistake that appeared after the revision. The terminology "multivariate model", is not correct in this situation (although I am aware that it is sometimes used) Please use the term multivariable instead (10.2105/AJPH.2012.300897).

We thank reviewer 1 for point this out and it is now updated to "multivariable".

Reviewer: 2

Competing interests of Reviewer: None